# SIMPLE OPTIMIZERS FOR CONVEX ALIGNED MULTI-OBJECTIVE OPTIMIZATION

## ABSTRACT

It is widely recognized in modern machine learning practice that access to a diverse set of tasks can enhance performance across those tasks. This observation suggests that, unlike in general multi-objective optimization, the objectives in many real-world settings may not be inherently conflicting. To address this, prior work introduced the Aligned Multi-Objective Optimization (AMOO) framework and proposed gradient-based algorithms with provable convergence guarantees. However, existing analysis relies on strong assumptions, particularly strong convexity, which implies the existence of a unique optimal solution. In this work, we relax this assumption and study gradient-descent algorithms for convex AMOO under standard smoothness or Lipschitz continuity conditions-assumptions more consistent with those used in deep learning practice. This generalization requires new analytical tools and metrics to characterize convergence in the convex AMOO setting. We develop such tools, propose scalable algorithms for convex AMOO, and establish their convergence guarantees. Additionally, we prove a novel lower bound that demonstrates the suboptimality of naive equal-weight approaches compared to our methods.

## 1 INTRODUCTION

Many real-world applications involve optimizing multiple objectives rather than focusing on a single objective. These problems have been extensively studied in the field of multi-objective optimization (MOO), by characterizing tradeoffs between competing objectives, namely, studying the Pareto front, or finding solutions that optimize particular desirable outcomes. In machine learning, MOO has gained significant traction over the past decades, especially in the paradigm of multi-task learning (Caruana, 1997; Teh et al., 2017; Sener and Koltun, 2018; Yu et al., 2020a; Liu et al., 2021; Navon et al., 2022; Achituve et al., 2024), or optimizing in the presence of proxy tasks (Bastani, 2021; Woodworth et al., 2023; Shen et al., 2023). These resulted with algorithmic advancements in computer vision (Yuan et al., 2012; Zhang et al., 2014; Liu et al., 2019), large language models (Uesato et al., 2022; Lightman et al., 2023; Yang et al., 2024; Wang et al., 2024; Guo et al., 2025; Team et al., 2025), and reinforcement learning (Jaderberg et al., 2016; Teh et al., 2017; Yu et al., 2020b; Veeriah et al., 2019; Dann et al., 2023; D'Eramo et al., 2024) when trained with multiple reward functions to name only a handful. For example, in computer vision tasks, researchers seek representations that jointly optimize different tasks such as depth estimation and segmentation. More recently, in LLM post-training, algorithm designers incorporate multiple reward functions during the post-training phase, optimizing for attributes like response length, and alignment to different reward functions.

These examples reveal a recurring theme: in many machine learning applications, algorithm designers supply multiple objectives that are not necessarily in conflict with each other, but rather help to better specify the desired solution. This leads to a natural question: Can we develop optimizers that benefit from multi-objective feedback when the objectives are aligned? Recently, Efroni et al. (2025) formalized the Aligned MOO (AMOO) setting, where the set of objectives is assumed to have a shared optimal solution, and designed gradient-descent algorithms for their framework. However, their work relied on several strict assumptions that often do not hold in practice. Arguably, the most restrictive one is requiring a variant of the strong convexity assumption—a condition rarely satisfied in practical deep learning applications–to hold. Such assumption implies the existence of a *unique* solution that optimizes all objective functions. Yet, in common deep learning practice, the loss

| Algorithm | Convergence of $\text{MG}(\bar{x})$ |
|---|---|
| `EW` | $\Omega(\sqrt{m}G/\sqrt{K})$ |
| `PAMOO` | $\mathcal{O}(G/\sqrt{K})$ |
| `MG-AMOO` w Polyak or GD | $\mathcal{O}(G/\sqrt{K})$ |
| `MG-AMOO` w Online Learner | $\mathcal{O}(\text{Regret}(K)/K)$ |

Table 1: Main results introduced in this work that characterize the convergence of the maximum gap metric, $\text{MG}(\bar{x})$. Here we provide the results for the $G$-Lipschitz function class; we include additional results for $\beta$-smooth functions in Section 5.2. $K$ is number of iterations, $m$ number of objectives, $\text{Regret}(K)$ represents the regret of an online learner used in the meta-algorithm we design.

surface tends to be flat, namely, have small curvature (Li et al., 2018; Petzka et al., 2021). Further, practitioners can directly optimize a sharpness aware loss term to reduce the flatness further (Foret et al., 2020). This suggest that in practice the strong convexity parameter may be arbitrarily small.

In this work, we develop provable and simple gradient-descent algorithms for AMOO under milder and more realistic assumptions compared to prior work. We avoid strong convexity assumptions, instead requiring only that functions be convex with either bounded Lipschitz constants or smoothness parameters. This is motivated by the fact that other common optimizers for single objective optimization in deep learning literature were developed for similar function classes (Zinkevich, 2003; Duchi et al., 2011; Kingma and Ba, 2014; Defazio et al., 2024). Removing the strong convexity assumption in the AMOO setting introduces new challenges. Since there is no unique optimal solution, we cannot measure convergence to a specific point $\mathbf{x}_\star$, unlike in prior work. Instead, we propose an alternative metric to characterize convergence in the convex AMOO setting: the Maximum Gap (MG), defined as $\text{MG}(\mathbf{x}) := \max_{i \in [m]} (f_i(\mathbf{x}) - \min_{\mathbf{x}} f_i(\mathbf{x}))$. Our goal is to achieve fast convergence to a solution where $\text{MG}(\mathbf{x}) = 0$. This metric is particularly appropriate for AMOO since in this framework functions can be optimized simultaneously and, hence, the MG should approach zero.

We begin by establishing a surprising lower bound for the naive Equal-Weight (EW) algorithm (Section 4), which applies GD to the sum of objective functions. Our analysis reveals that the MG metric may converge with polynomial dependence on the number of functions—an undesirable outcome implying that convergence deteriorates when incorporating additional loss functions. We then develop specialized Gradient-Descent (GD) methods for the AMOO setting that achieve faster convergence rates for the MG metric, independent of the number of functions (Section 5). We provide a new analysis of the `PAMOO` algorithm, and then design `MG-AMOO`, a meta-algorithm that uses only a single-objective optimizer to solve an AMOO problem. Instantiating it with an online learner, Polyak step-size design or GD algorithms leads to new, scalable and simple algorithms for AMOO (see Table 1). To the best of our knowledge, this represents the first non-asymptotic provable advantage of algorithms specifically designed for AMOO compared to the naive EW method. We conclude by conducting experiments with the different algorithms we introduced and comparing their performance to the naive EW as well as comparing their running time (Section 7).

## 2 PRELIMINARIES: ALIGNED MULTI OBJECTIVE OPTIMIZATION

We describe the AMOO setting and the structural assumptions used in this work. Assume access to multi-objective feedback in the form of a vector-valued function $F : \mathbb{R}^n \to \mathbb{R}^m$, $F(\mathbf{x}) = (f_1(\mathbf{x}), f_2(\mathbf{x}), \ldots, f_m(\mathbf{x}))$, and the optimal values of the individual objectives is denoted by $f_i^\star := \min_{\mathbf{x} \in \mathbb{R}^n} f_i(\mathbf{x})$ for $i \in [m]$. Given a weight vector $\mathbf{w} \in \mathbb{R}_+^m$ we let $f_{\mathbf{w}}(\mathbf{x}) := \mathbf{w}^T F(\mathbf{x})$ be the weighted function by $\mathbf{w}$. In the AMOO setting, we further assume that the objectives are *aligned*, in the sense that they can be minimized simultaneously. That is, there exists a non-empty set of points $C_\star = \{\mathbf{x} \in \mathbb{R}^n | f_i(\mathbf{x}) = f_i^\star, \ \forall i \in [m]\}$ that minimizes all functions $\{f_i(\mathbf{x})\}_{i \in [m]}$. Namely, for any $\mathbf{x}_\star \in C_\star$ it holds that

$$\mathbf{x}_\star \in \arg \min_{\mathbf{x} \in \mathbb{R}^n} f_i(\mathbf{x}) \quad \forall i \in [m].$$

In this work we consider the convex setting, where each function $f_i$ is convex. We also assume standard structural properties of the objectives smoothness and Lipschitz continuity, which are

commonly made in the literature (Zinkevich, 2003; Duchi et al., 2011; Kingma and Ba, 2014; Defazio et al., 2024). Let $\|\cdot\|$ denote the Euclidean $L_2$ norm. A function is called $G$-Lipschitz if for all $\mathbf{x} \in \mathbb{R}^n$, it holds that $\|\nabla f(\mathbf{x})\| \leq G$. A function $f : \mathbb{R}^n \to \mathbb{R}$ is said to be $\beta$-smooth if for all $\mathbf{x}, \mathbf{y} \in \mathbb{R}^n$: $f(\mathbf{y}) \leq f(\mathbf{x}) + \nabla f(\mathbf{x})^\top (\mathbf{y} - \mathbf{x}) + \frac{\beta}{2} \|\mathbf{x} - \mathbf{y}\|^2$. Every $\beta$-smooth function has the following useful consequence.

**Lemma 1** (Standard result, E.g., 9.17 Boyd and Vandenberghe (2004)). *Let $f : \mathbb{R}^n \to \mathbb{R}$ a $\beta$-smooth over $\mathbb{R}^n$, and let $f^\star := \min_{\mathbf{x} \in \mathbb{R}^n} f(\mathbf{x})$. Then, for every $\mathbf{x} \in \mathbb{R}^n$ it holds that $\|\nabla f(\mathbf{x})\|^2 \leq 2\beta \left( f(\mathbf{x}) - f^\star \right)$.*

**Additional notations:** We denote by $\Delta_m$ as the m-dimensional simplex and by $\mathbf{e}_i$ as the one-hot vector on the $i^{th}$ index, $\mathbf{e}_{i,j} = 1$ if $i = j$ and $\mathbf{e}_{i,j} = 0$ otherwise.

## 3 PRIOR WORK

Multi-objective optimization (Gunantara, 2018) has a rich research history and found applications in many different domains. In machine learning applications, algorithm designers often apply such paradigm when few sources of feedback are available in multi-task learning or in the presence of proxy loss functions. From optimization and machine learning perspectives, most of previous research efforts have been devoted to approximating the Pareto front of optimal solutions or finding good balance between the different objectives. In this work, we focus on the AMOO setting and aim to develop improved algorithms under the alignment assumption.

The AMOO setting that we consider in this work was introduced by Efroni et al. (2025). Additionally on making the alignment assumption between the multiple objectives, the authors required a strong convexity parameter $\mu_G := \min_{\mathbf{x}} \max_{\mathbf{w} \in \Delta_m} \lambda_{\min} \left( \nabla^2 f_{\mathbf{w}}(\mathbf{x}) \right)$ to be positive $\mu_G > 0$. Such assumption also implies there exist a unique $\mathbf{x}_\star$ that optimizes $\{f_i\}_{i \in [m]}$ which enables to measure converence of algorithms by measuring their convergence to $\mathbf{x}_\star$. The authors designed the CAMOO and PAMOO algorithms and established both converge in linear rate to $\mathbf{x}_\star$ in the parameter $\mu_G/\beta$.

Extending their result to the convex setting requires new analysis and perspective on AMOO. First, CAMOO is not a valid approach if $\mu_G = 0$. CAMOO chooses the weighting vector in the $k^{\text{th}}$ iteration as $\mathbf{w}_k \in \arg\max_{\mathbf{w} \in \Delta_m} \lambda_{\min} \left( \nabla^2 f_{\mathbf{w}}(\mathbf{x}_k) \right)$, however, for the non-strongly convex setting, such approach is expected to fail. For general convex functions the solution of such optimization problem can be arbitrary: we can find a set of convex functions for which for any weight vector $\mathbf{w}$ it holds that $\lambda_{\min} \left( \nabla^2 f_{\mathbf{w}}(\mathbf{x}) \right) = 0$, and, hence, for such example, the solution can be the entire simplex (see Appendix B.1 for an example). This suggests CAMOO is flawed for the non-strongly convex setting. Additionally, and more crucially, in the convex setting, without strong convexity assumption, it does not hold that there exists a *unique* $\mathbf{x}_\star$. Hence, establishing convergence of $\|\mathbf{x}_\star - \mathbf{x}_k\|$ is no longer possible, and one is required to rethink of the metrics to characterize convergence. In this work, we provide an analysis of PAMOO while removing the strong convexity assumption, as well as introducing a computationally cheap meta-algorithm for the AMOO framework.

## 4 THE MAXIMUM-GAP METRIC & A LOWER BOUND

Without the strong convexity assumption, we can no longer establish point-wise convergence guarantees. What metric should we use in this scenario? Since all objectives can be jointly optimized, we measure convergence via the *Maximum Gap (MG)* across the different objectives, defined as:

$$\text{MG}(\mathbf{x}) := \max_{i \in [m]} \left( f_i(\mathbf{x}) - f_i^\star \right).$$

The MG metric characterizes the worst-case convergence across all functions; as such, it captures a robustness aspect of the solution. This criterion is also explored in the field of robust optimization Ben-Tal et al. (2009), and typically leads to more challenging optimization problems Ben-Tal and Nemirovski (2002); Wiesemann et al. (2013); Chen et al. (2017). However, under the alignment or approximate alignment assumptions, we may hope to obtain convergence guarantees with simple GD algorithms with respect to this robust criterion. Indeed, under such assumptions, we expect the MG to become negligible as the algorithm converges—namely, $\lim_{k \to \infty} \text{MG}(\mathbf{x}_k) \sim 0$. Importantly,

this should hold without requiring strong convexity, since convergence is measured with respect to the optimal values, as opposed to point-wise convergence to a single optimal solution.

Next, we focus on the naive Equal-Weights (EW) baseline, which applies a GD algorithm to a single objective that uniformly weights all component objectives:

$$f_{\text{EW}}(\mathbf{x}) = \frac{1}{m} \sum_{i \in [m]} f_i(\mathbf{x}).$$

This approach is arguably the simplest baseline for multi-objective and multi-task problems (Teh et al., 2017; Sener and Koltun, 2018; Yu et al., 2020a). Notably, the EW baseline also produces a solution that minimizes the MG metric, as the following simple claim shows (see Appendix B.2).

**Claim 1.** *Assume* $\{f_i\}_{i \in [m]}$ *are aligned. Let* $\mathbf{x}_{\text{EW}}^\star \in \arg\min_{\mathbf{x} \in \mathbb{R}^n} f_{\text{EW}}(\mathbf{x})$. *Then,* $\text{MG}(\mathbf{x}_{\text{EW}}^\star) = 0$.

Although EW optimizes the MG metric asymptotically, one may ask how the convergence rate is effected by the number of objective, and whether it can be improved by alternative algorithms. In what follows, we provide a lower bound showing that, for EW, the MG metric may converge with polynomial dependence on the number of functions. This indicates a significant disadvantage of the EW approach. We focus on a variant of the method that selects the step size via the Polyak step-size design which is adaptively chooses the learning rate Polyak (1987); Hazan and Kakade (2019). The gradient-based algorithm outputs $\bar{\mathbf{x}} = \frac{1}{K} \sum_{k=1}^{K} \mathbf{x}_k$, where the iterates are generated by:

$$\mathbf{x}_{k+1} = \mathbf{x}_k - \eta_k \nabla f_{\text{EW}}(\mathbf{x}_k), \text{ with } \eta_k = \frac{f_{\text{EW}}(\mathbf{x}_k) - f_{\text{EW}}(\mathbf{x}_{\text{EW}}^\star)}{\|\nabla f_{\text{EW}}(\mathbf{x}_k)\|^2}. \tag{1}$$

**Theorem 1** (Lower Bound for EW with Polyak Step Size). *There exists a set of functions* $\{f_i(\mathbf{x})\}_{i \in [m]}$ *that are G-Lipschitz and convex such that, for* $K \leq m$ *iterations, the worst-case gap that Equal Weights with the Polyak algorithm guarantees depends polynomially on* $m$. *Specially, the output of GD with Polyak step size converges with rate* $\text{MG}(\bar{\mathbf{x}}) \in \Omega\left(\sqrt{m}G \|\mathbf{x}_1 - \mathbf{x}_{\text{EW}}^\star\| / \sqrt{K}\right)$.

*Proof of Theorem 1.* Let $\epsilon > 0$, $n \geq m > 1$, and $a = (m-1)\epsilon$. Let $f_i(\mathbf{x}) = |x_i|$ for every $i \in [m]$, which are 1-Lipschitz ($G = 1$), and convex functions, and initial point $\mathbf{x}_1 = (x_1, x_2, \ldots, x_n) = (a, \epsilon, \ldots, \epsilon, 0, \ldots, 0) \in \mathbb{R}^n$, i.e. $x_1 = a$, $x_i = \epsilon$ for every $i \in [2, m]$, and $x_i = 0$ for every $i \in [m+1, n]$. Note that, the subgradient is $\nabla f_i(\mathbf{x}) = \text{sign}(x_i)$ for every $i \in [m]$ and for every point that $x_i \neq 0$. When $x_i = 0$ the subgradient is $\nabla f_i(\mathbf{x}) \in [-1, 1]$.

Observe that, from equation 1, the update rule is given by

$$\mathbf{x}_{k+1} = \mathbf{x}_k - \frac{\sum_i f_i(\mathbf{x}_k) - f_i^\star}{\|\sum_i \nabla f_i(\mathbf{x}_k)\|^2} \sum_i \nabla f_i(\mathbf{x}_k), \tag{2}$$

since

$$\eta_k \nabla f_{\text{EW}}(\mathbf{x}_k) = \frac{m^{-1}\sum_i f_i(\mathbf{x}_k) - f_i^\star}{\|m^{-1}\sum_i \nabla f_i(\mathbf{x}_k)\|^2} m^{-1} \sum_i \nabla f_i(\mathbf{x}_k) = \frac{\sum_i f_i(\mathbf{x}_k) - f_i^\star}{\|\sum_i \nabla f_i(\mathbf{x}_k)\|^2} \sum_i \nabla f_i(\mathbf{x}_k).$$

From equation 2 The update step is:

$$\mathbf{x}_{k+1} = \mathbf{x}_k - \frac{\sum_{i \in [m]} |x_{k,i}|}{m} \left(\text{sign}(x_{k,1}), \ldots, \text{sign}(x_{k,m}), 0, \ldots, 0\right).$$

Using Lemma 3 (see Appendix C.1) it holds that after $k$ iterations the first component of the iterate, denoted by $x_{k,1}$ is:

$$x_{k,1} = a\left(1 - \frac{2}{m}\right)^{k-1}.$$

Recall that the return point is $\bar{\mathbf{x}} = \frac{1}{K} \sum_{k=1}^{K} \mathbf{x}_k$. Which holds that the first component of it is:

$$\bar{x}_1 = \frac{a}{K} \sum_{k=1}^{K} \left(1 - \frac{2}{m}\right)^{k-1} = \frac{a}{K} \frac{1 - \left(1 - \frac{2}{m}\right)^K}{1 - \left(1 - \frac{2}{m}\right)} = \frac{a}{K} \frac{1 - \left(1 - \frac{2}{m}\right)^K}{\frac{2}{m}} = \frac{ma}{2K}\left(1 - \left(1 - \frac{2}{m}\right)^K\right).$$

---

**Algorithm 1** `PAMOO`

> **initialize:** $\{f_i^\star\}_{i\in[m]}, K$.
> **for** $k = 1, 2, \ldots, K$ **do**
> $\quad \Delta_k := [f_1(\mathbf{x}_k) - f_1^\star, \ldots, f_m(\mathbf{x}_k) - f_m^\star]$
> $\quad \mathbf{J}_k := [\nabla f_1(\mathbf{x}_k) \ldots \nabla f_m(\mathbf{x}_k)] \in \mathbb{R}^{n \times m}$
> $\quad \mathbf{w}_k \in \arg\max_{\mathbf{w}\in\mathbb{R}_+^m} 2\mathbf{w}^\top \Delta_k - \mathbf{w}^\top \mathbf{J}_k^\top \mathbf{J}_k \mathbf{w}$
> $\quad \mathbf{x}_{k+1} = \mathbf{x}_k - \nabla f_{\mathbf{w}_k}(\mathbf{x}_k)$
> **end for**
> **return:** $\bar{\mathbf{x}} = \frac{1}{K} \sum_{k=1}^K \mathbf{x}_k$

---

**Algorithm 2** `MG-AMOO`

> 1: **initialize:** $\{f_i^\star\}_{i\in[m]}, K,$ `SOO`
> 2: **for** $k = 1, 2, \ldots, K$ **do**
> 3: $\quad I(k) \in \arg\max_{i\in[m]} f_i(\mathbf{x}_k) - f_i^\star$
> 4: $\quad \mathbf{x}_{k+1} = \text{SOO}(f_{I(k)}, x_k)$
> 5: **end for**
> 6: **return:** $\bar{\mathbf{x}} = \frac{1}{K} \sum_{k=1}^K \mathbf{x}_k$

---

By using equation 8, it holds that

$$\bar{x}_1 = \frac{ma}{2K}\left(1 - \left(1 - \frac{2}{m}\right)^K\right) \geq \frac{ma}{2K}\frac{2K}{m+2K} = \frac{ma}{m+2K}.$$

Recall that $a > 0$, and $K > 0$. Then the RHS of the last inequality, i.e. $\frac{ma}{m+2K}$, is a decreasing function by $K$. Thus, for $K \leq m$ iterations we have that

$$\max_{i\in[m]} f_i(\bar{\mathbf{x}}) - f_i^\star \geq f_1(\bar{\mathbf{x}}) - f_1^\star = |\bar{x}_1| - 0 \geq \frac{ma}{m+2K} \geq \frac{\sqrt{m}\sqrt{K}a}{3K} = \frac{\sqrt{m}a}{3\sqrt{K}}.$$

Since $\mathbf{x}_{\text{EW}}^\star = \mathbf{0}$, it holds that $\|\mathbf{x}_1 - \mathbf{x}_{\text{EW}}^\star\| = \sqrt{a^2 + (m-1)\epsilon^2} = \sqrt{a^2 + \frac{a^2}{m-1}} = a\sqrt{\frac{m}{m-1}}$. Then, we obtain that the max-gap is lower bounded by:

$$\text{MG}(\bar{\mathbf{x}}) = \max_{i\in[m]} f_i(\bar{\mathbf{x}}) - f_i^\star \geq \frac{\sqrt{m}a}{3\sqrt{K}} = \frac{\sqrt{m-1}\|\mathbf{x}_1 - \mathbf{x}_{\text{EW}}^\star\|}{3\sqrt{K}} = \sqrt{\frac{m-1}{9}}\frac{G\|\mathbf{x}_1 - \mathbf{x}_{\text{EW}}^\star\|}{\sqrt{K}}.$$

$\square$

## 5 SIMPLE OPTIMIZERS FOR CONVEX AMOO

In the previous section, we introduced the MG metric as a meaningful objective for optimization in the general convex AMOO setting, without relying on strong convexity assumptions. We also established a lower bound for the `EW` algorithm, showing that the MG may converge at a polynomial rate in the number of objective functions. In this section, we study specialized GD methods for AMOO and establish their convergence rates are independent of the number of objectives. First, in Section 5.1, we provide a new analysis of the Polyak Aligned Multi-Objective Optimizer (`PAMOO`). Then, in Section 5.2, we introduce an algorithmic reduction from AMOO to single-objective optimization. Instantiating this reduction with standard optimization methods yields simple and efficient algorithms for AMOO, which enjoys best of both worlds. We conclude by comparing the computational complexity of existing approaches in Section 5.3 and show that the new `MG-AMOO` method is significantly more efficient than `PAMOO` and comparable in cost to the `EW` baseline.

### 5.1 POLYAK METHOD FOR AMOO (`PAMOO`)

The Polyak step-size design Polyak (1987); Hazan and Kakade (2019) provides a parameter-free way to choose the learning rate in single-objective optimization problems. This approach was generalized to the AMOO setting in Efroni et al. (2025), resulting in the `PAMOO` algorithm (Algorithm 1). Its derivation is motivated by arguments similar to those behind the original Polyak step-size.

Assume the update rule of $\mathbf{x}_k$ has the form $\mathbf{x}_{k+1} = \mathbf{x}_k - \nabla f_{\mathbf{w}}(\mathbf{x}_k)$ for a fixed positive weight vector $\mathbf{w} \in \mathbb{R}_+^m$. Then, due to the convexity of $f_{\mathbf{w}}$, the following inequality holds:

$$\|\mathbf{x}_{k+1} - \mathbf{x}\|^2 \leq \|\mathbf{x}_k - \mathbf{x}\|^2 - 2\left(f_{\mathbf{w}}(\mathbf{x}_k) - f_{\mathbf{w}}(\mathbf{x})\right) + \|\nabla f_{\mathbf{w}}(\mathbf{x}_k)\|^2. \quad (3)$$

Choosing $\mathbf{w}_k \in \arg\max_{\mathbf{w}\in\mathbb{R}_+^m} 2\left(f_{\mathbf{w}}(\mathbf{x}_k) - f_{\mathbf{w}}(\mathbf{x})\right) - \|\nabla f_{\mathbf{w}}(\mathbf{x}_k)\|^2$ to maximize the decrease results in the `PAMOO` method.

Prior work established convergence of `PAMOO` under strong convexity and self-concordance assumptions, which often do not hold in practical settings. The following result, Theorem 2, establishes convergence of `PAMOO` in the convex setting, removing the previously strict assumptions. Moreover, unlike `EW`, the convergence of `PAMOO` is independent of the number of objective functions. The proof of the theorem appears in Appendix E.

**Theorem 2** (Convergence of `PAMOO`). *Algorithm 1 guarantees that after $K$ iterations, the worst-case gap is upper bounded by:*

(1) $\quad \max_{i \in [m]} f_i(\bar{\mathbf{x}}) - f_i^\star \leq G \|\mathbf{x}_1 - \mathbf{x}_\star\| / \sqrt{K}, \quad$ *when* $\{f_i(\mathbf{x})\}_{i \in [m]}$ *are $G$-Lipschitz.*

(2) $\quad \max_{i \in [m]} f_i(\bar{\mathbf{x}}) - f_i^\star \leq 2\beta \|\mathbf{x}_1 - \mathbf{x}_\star\| / K, \quad$ *when* $\{f_i(\mathbf{x})\}_{i \in [m]}$ *are $\beta$-smooth.*

This result establishes convergence of the MG metric at a rate that is independent of the number of objective functions, representing a substantial improvement over the `EW` method, which converges at a rate polynomial in the number of objectives (Theorem 1).

### 5.2 REDUCTION APPROACH: MAXIMUM-GAP AMOO (`MG-AMOO`)

In this section, we introduce a meta-algorithm for the AMOO setting that uses a single-objective optimizer in a black-box manner and returns a near-optimal solution for the AMOO problem. Specifically, we design a reduction from AMOO to single-objective optimization. We show that any online learning algorithm with vanishing regret can be used to solve an AMOO instance.

We refer to this algorithmic reduction as Maximum Gap AMOO (`MG-AMOO`), see Algorithm 2. The algorithm assumes access to a single-objective optimizer (`SOO`)—a procedure that receives the current iterate $\mathbf{x}_k$, a function $f$, and returns an updated iterate $\mathbf{x}_{k+1}$. Similar to `PAMOO`, it also assumes access to the optimal values $\{f_i^\star\}_{i \in [m]}$. At each iteration, the function with the largest gap is selected, and `SOO` is called to produce the next iterate.

This reduction is motivated by the following simple lemma:

**Lemma 2** (Reduction: AMOO to Single-Objective Optimization). *Let $I(k) \in \arg\max_{i \in [m]} f_i(\mathbf{x}_k) - f_i^\star$, and let $\{\mathbf{x}_k\}_{k=1}^K$ be the optimization trajectory. Any algorithm that returns the average iterate $\bar{\mathbf{x}} = \frac{1}{K} \sum_{k=1}^K \mathbf{x}_k$ guarantees the worst-case max-gap is upper bounded by*

$$\text{MG}(\bar{\mathbf{x}}) := \max_{i \in [m]} f_i(\bar{\mathbf{x}}) - f_i^\star \leq \frac{1}{K} \sum_{k=1}^K \left( f_{I(k)}(\mathbf{x}_k) - f_{I(k)}^\star \right). \tag{4}$$

*Proof of Lemma 2.* Denote $I_e \in \arg\max_{i \in [m]} f_i(\bar{\mathbf{x}}) - f_i^\star$, the index of the function with the maximum gap at $\bar{\mathbf{x}}$. Since $I(k) \in \arg\max_{i \in [m]} f_i(\mathbf{x}_k) - f_i^\star$, it follows that for every $k$:

$$f_{I_e}(\mathbf{x}_k) - f_{I_e}^\star \leq f_{I(k)}(\mathbf{x}_k) - f_{I(k)}^\star.$$

By convexity of $\{f_i\}_{i \in [m]}$ and Jensen's inequality, we have $f_{I_e}(\bar{\mathbf{x}}) \leq \frac{1}{K} \sum_{k=1}^K f_{I_e}(\mathbf{x}_k)$. Summing over $k$:

$$\max_{i \in [m]} f_i(\bar{\mathbf{x}}) - f_i^\star = f_{I_e}(\bar{\mathbf{x}}) - f_{I_e}^\star \leq \frac{1}{K} \sum_{k=1}^K \left( f_{I_e}(\mathbf{x}_k) - f_{I_e}^\star \right) \leq \frac{1}{K} \sum_{k=1}^K \left( f_{I(k)}(\mathbf{x}_k) - f_{I(k)}^\star \right).$$

$\square$

This result implies that we can minimize the MG of the average iterate by generating a sequence of iterates with low average regret with respect to the functions selected by `MG-AMOO`, which always chooses the function with the largest current gap.

A direct corollary of Lemma 2 is a reduction from AMOO to online learning (Cesa-Bianchi and Lugosi, 2006; Hazan et al., 2016; Orabona, 2019). The online learning literature provides algorithms—such as online GD, online mirror descent, and online Newton step—that achieve sublinear regret on sequences of adaptive adversarially chosen functions (Hazan et al., 2016):

$$\text{Regret}(K) := \sup_{\{f_1, \ldots, f_K\} \subseteq \mathcal{F}} \left( \sum_{k=1}^K f_k(\mathbf{x}_k) - \sum_{k=1}^K f_k(\mathbf{x}) \right) \in o(K), \tag{5}$$

for any bounded convex functions, $\{f_k\}_{k=1}^{K}$, over $\mathcal{X}$, and for any $\mathbf{x} \in \mathcal{X}$, when $\mathcal{X}$ is a bounded convex set.

Instantiating MG-AMOO with such an online learner yields the following guarantee:

**Corollary 1** (Online to Aligned Multi-Objective (O2AMO))**.** *Let* $\mathcal{C}_\star$ *a bounded set. Let* $\{f_i\}_{i \in [m]}$ *satisfy the AMOO assumption and be bounded over* $\mathcal{X} := ConvexHull\big\{\{\mathbf{x}_1, \ldots, \mathbf{x}_K\} \bigcup \mathcal{C}_\star\big\}$. *Instantiating the single-objective optimizer* SOO *in* MG-AMOO *with an adaptive adversary online learning algorithm with regret* $\mathrm{Regret}(K)$, *guarantees that* $MG(\bar{\mathbf{x}}) \leq \mathrm{Regret}(K)/K$.

*Proof of Corollary 1.* Recall that $\{f_{I(k)}\}_{k=1}^{K} \subseteq \{f_i\}_{i=1}^{m}$ are bounded and convex over $\mathcal{X}$, which is bounded and convex itself. Then, by the regret guarantee in equation 5, it holds for any $\mathbf{x} \in \mathcal{X}$ that $\sum_{k=1}^{K} f_{I(k)}(\mathbf{x}_k) - f_{I(k)}(\mathbf{x}) \leq \mathrm{Regret}(K) \in o(K)$, where $I(k)$ is chosen adversarially based on $x_k$ (see Algorithm 2). Setting $\mathbf{x} = \mathbf{x}_\star \in \mathcal{C}_\star \subseteq \mathcal{X}$ and applying Lemma 2 completes the proof. $\square$

This guarantee is analgous to the well-known Online-to-Batch (O2B) conversion. In words, if an online learner achieves sublinear regret, then averaging its iterates over a data sequence yields strong convergence guarantees in the MG-AMOO setting.

We also analyze two additional and widely used single-objective optimization algorithms—GD and the Polyak step-size design method—and show that both can be applied in the smooth and Lipchitz settings. The Polyak step-size design and GD in the smooth case require different specialized analysis to bound the term in the right hand side of equation 4. The analysis of these algorithms and settings is not included in standard online learning analysis. The result for GD in the Lipchitz case is a direct application of standard online learning analysis.

**Theorem 3** (MG-AMOO: Gradient-Descent and Polyak step size method)**.** *Assume that* Polyak *or* GD *methods are used as the* SOO. MG-AMOO *(Algorithm 2) guarantees that after* $K$ *iterations the worst-case gap is upper bounded by:*

(1) $\quad \mathrm{MG}(\bar{\mathbf{x}}) := \max_{i \in [m]} f_i(\bar{\mathbf{x}}) - f_i^\star \leq \dfrac{3}{2} \dfrac{G \, \|\mathbf{x}_1 - \mathbf{x}_\star\|}{\sqrt{K}}, \quad$ *when* $\{f_i(\mathbf{x})\}_{i \in [m]}$ *are* $G$-*Lipschitz.*

(2) $\quad \mathrm{MG}(\bar{\mathbf{x}}) := \max_{i \in [m]} f_i(\bar{\mathbf{x}}) - f_i^\star \leq \dfrac{2\beta \, \|\mathbf{x}_1 - \mathbf{x}_\star\|}{K}, \quad$ *when* $\{f_i(\mathbf{x})\}_{i \in [m]}$ *are* $\beta$-*smooth.*

The proof of the theorem is in Appendix F.

## 5.3 Computational Complexity

Among the three algorithmic approaches considered in this work, PAMOO requires the highest computational cost. Specifically, computing the Jacobian incurs a cost of $O(nm^2)$ operations, where $n$ is the dimension of the parameter space and $m$ is the number of objective functions. Additionally, each iteration requires solving a quadratic constrained optimization problem. Although certain techniques may reduce the overall burden, the need to compute gradients with respect to all functions and solve a convex optimization problem at each step introduces significant computational and implementation complexity.

In contrast, both the EW and MG-AMOO methods require only $O(m)$ operations per iteration: the former to sum the functions, and the latter to identify the function with the maximum gap. The overall computational cost of these methods also depends on the complexity of the single-objective optimizer (SOO) employed. Since both EW and MG-AMOO can leverage simple optimizers such as GD or projected GD, they are often substantially cheaper to apply in practice.

## 6 Relaxing AMOO Assumption: $\epsilon$-Approximate AMOO

The alignment assumption requires that all objective functions share a common optimal solution. However, in practice, we may only expect the existence of a near-optimal solution that approximately minimizes all functions. To address such scenarios, we seek algorithms that are robust to this relaxation of the alignment setting.

| Algorithm | SGD [iter/time] ↑ | Adam [iter/time] ↑ |
|---|---|---|
| EW | 1 | 1 |
| PAMOO | 0.07 | 0.08 |
| MG-AMOO w/ Polyak | 0.97 | 0.96 |
| MG-AMOO w/ GD | 0.84 | 0.85 |

Table 2: Average iterations per time across problems (P1)-(P3) relative to EW. MG-AMOO and EW running time are substantially better compared to PAMOO.

To formalize this approximate scenario, we define the set of $\epsilon$-approximate solutions:

$$C_\epsilon = \{\mathbf{x} \in \mathbb{R}^n \mid f_i(\mathbf{x}) - f_i^\star \leq \epsilon \quad \forall i \in [m]\}. \tag{6}$$

We assume that $C_\epsilon$ is non-empty and contains at least one element. This assumption generalizes the alignment condition in AMOO; indeed, setting $\epsilon = 0$ recovers the original AMOO setting.

The analytical tools developed in this work extend naturally to the $\epsilon$-approximate AMOO setting. In particular, we show that all the proposed algorithms remain applicable, recovering solutions that are $\epsilon$-near-optimal across all objectives. Below we provide an informal statement of the result, and refer the reader to the appendix for complete proofs.

**Theorem 4** ((Informal) Convergence in Convex $\epsilon$-Approximate AMOO). *Let $\epsilon > 0$ and assume $C_\epsilon$ is non-empty. Then, PAMOO and MG-AMOO, instantiated with an online learner, GD, or the Polyak step-size method, output a solution $\bar{\mathbf{x}}$ satisfying asymptotically* $\mathrm{MG}(\bar{\mathbf{x}}) \leq \epsilon$.

This result implies that both PAMOO and MG-AMOO are applicable in the $\epsilon$-approximate AMOO setting. Notably, this substantially improves upon the results of Efroni et al. (2025): their results are applicable in the approximate case only if $\epsilon \leq \epsilon_0$, with $\epsilon_0$ depending on structural parameters such as smoothness and strong convexity. In contrast, our analysis removes this restriction and establishes that both PAMOO and MG-AMOO can recover an $\epsilon$-near-optimal solution for any value of $\epsilon$.

## 7 EXPERIMENTS

In this section, we tested the algorithms EW, PAMOO and MG-AMOO the reduction approach we introduced in this work. We experimented with an additional variation of MG-AMOO with weight averaging or momentum, as we refer to it. That is, we calculate the new weight vector $\mathbf{w}_{\mathrm{new}}$ according to the MG-AMOO update rule and update the weight by $\mathbf{w}_{k+1} = (1 - \beta_{\mathrm{mom}})\mathbf{w}_k + \beta_{\mathrm{mom}}\mathbf{w}_{\mathrm{new}}$. In all experiments, we set $\beta_{\mathrm{mom}} = 0.95$ similar to the default setting of other optimizers.

Similar to Efroni et al. (2025), in the learning problem we address, one network is tasked with replicating the outputs of another fixed network. The fixed network, characterized by parameters $\theta_\star$, is represented as $h_{\theta_\star} : \mathbb{R}^{d_i} \to \mathbb{R}^{d_o}$. The second network, with parameters $\theta$, is denoted as $h_\theta : \mathbb{R}^{d_i} \to \mathbb{R}^{d_o}$. Both networks are 2-layer neural networks utilizing ReLU activation functions and consist of 512 hidden units.

We draw data from a uniform distribution $\mathcal{D} = \{\mathbf{x}_i\}_i$ where $\mathbf{x}_i \in \mathrm{Uniform}([-1,1]^{d_i})$. We consider three loss functions:

$$\forall i \in [3]: \quad f_i(\theta) = \frac{1}{|\mathcal{D}|}\sum_{\mathbf{x} \in \mathcal{D}}\big((h_\theta(\mathbf{x}) - h_{\theta_\star}(\mathbf{x}) - \epsilon_i)^\top \mathbf{H}_i(h_\theta(\mathbf{x}) - h_{\theta_\star}(\mathbf{x})) - \epsilon_i\big)^{\alpha_i},$$

where $\mathbf{H}_i \in \mathbb{R}^{d_o \times d_o}$ is a positive definite matrix and $\alpha_i \geq 1$.

We choose 3 optimization problems (P1)-(P3) that demonstrate the behavior and robustness of our method, problems become progressively harder. Problem (P1) is the switching example of Efroni et al. (2025). We set $\mathbf{H}_i = \mathbf{I}$, and $\alpha_i \in \{1, 1.5, 2\}$ and $\epsilon_i = 0$. This implies the alignment assumption is satisfied. For such a choice $f_1$ has larger curvature for large losses whereas $f_2$ and $f_3$ have larger curvature for small losses. Hence, we expect the the multi-objective feedback can allow the optimizer to focus on different functions in different parts of the feature space. We demonstrate the switching of weights empirically in Appendix A. Problems (P2) is similar to

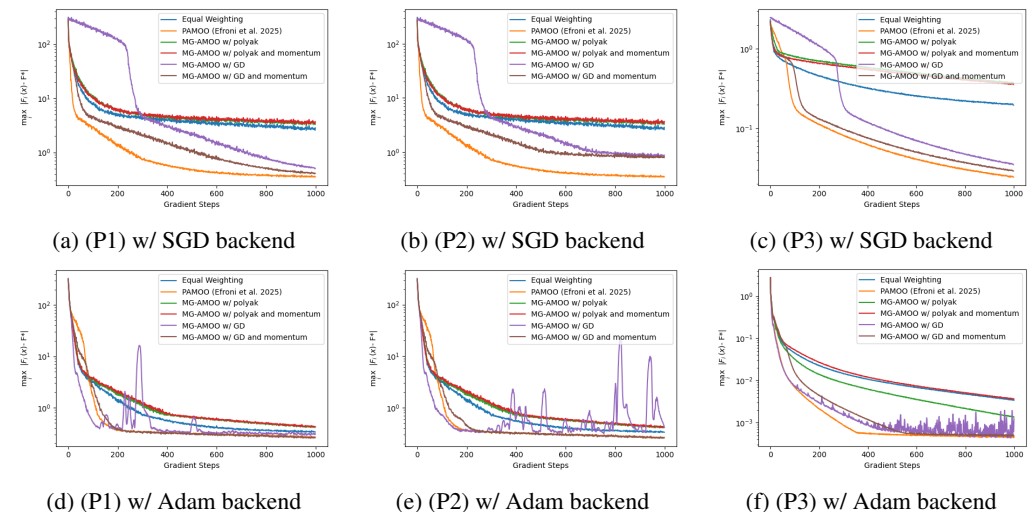

(a) (P1) w/ SGD backend     (b) (P2) w/ SGD backend     (c) (P3) w/ SGD backend

(d) (P1) w/ Adam backend    (e) (P2) w/ Adam backend    (f) (P3) w/ Adam backend

Figure 1: Performance of different AMOO algorithms with SGD (top) and Adam (bottom) backend. We show the convergence properties in three increasingly difficult problems: our switching problem (left), the switching problem when the optimum is not exactly alighned (middle), and when the problem is additionally ill-conditioned (right). When we augment our algorithm with momentum it can almost recover the performance of PAMOO at a fraction of the computational cost to run it.

(P1), while setting $\epsilon_i \in \{0, 0.05, -0.05\}$, which tests the setting in which the functions are not perfectly aligned. Problem (P3) additionally tests the performance of the algorithms for scenario $f_1, f_2$ and $f_3$ are not strongly convex in the network outputs, namely, when the Hessian is ill-conditioned. We implement this by setting the problem dimension $d_i = 100$, the Hessian to $\mathbf{H}_i = 0.5 \cdot \text{Diag}(u_1 + 1, \ldots, (u_{90} + 1)10^{-3}, \ldots, (u_{100} + 1)10^{-3})$, with $\epsilon_i \in \{0, 0.01, -0.01\}$. Our experiment is inspired by problem P1 tested in (Schneider et al., 2019; Chaudhari et al., 2019).

We used EW, PAMOO and MG-AMOO as the algorithms that choose the weight vector at each iteration $\mathbf{w}_k$, and applied SGD or ADAM optimizers to the weighted loss by $\mathbf{w}_k$. Figure 1 depicts the convergence of the algorithms. As observed, PAMOO, which requires more compute compared to MG-AMOO and EW achieves the best performance, and MG-AMOO is substantially better than EW, while suffering with no degradation in its running time (see Table 2). Additionally, it is evident the algorithms are also robust to small violations in the alignment assumption: the algorithms consistently converge on problems (P2) for instances of the $\epsilon$-approximate AMOO setting.

## 8   CONCLUSIONS, LIMITATIONS & FUTURE WORK

We studied the convex AMOO setting, introducing the MG metric and new analytical tools that remove the need for strong convexity. We proposed MG-AMOO, a simple meta-algorithm that leverages standard single-objective optimizers, and established convergence guarantees for its variants, as well as for the more compute-intensive PAMOO. We also derived a non-asymptotic lower bound for EW, showing a suprising disadvantage of its: its performance may degrade as the number of objectives increases. Additionally, we showed that both MG-AMOO and PAMOO remain robust even when the alignment assumption is relaxed. Experiments confirm that MG-AMOO achieves faster convergence than EW without incurring the high computational cost of PAMOO.

Future work can address the limitations of this study. Neither our work nor prior efforts have explored the stochastic or non-convex AMOOO settings, which are important for practical applications. Further research should also include large-scale experiments on computer vision, reinforcement learning, or LLMs with multiple reward signals. We conjecture that AMOO algorithms may benefit general multi-objective optimization by first identifying a strong common solution, followed by fine-tuning to satisfy specific trade-offs. This approach offers a new algorithmic perspective with potential impact on a wide range of machine learning applications in the future.

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
