# OpenReview forum: "Simple Optimizers for Convex Aligned Multi-Objective Optimization"
_ICLR.cc/2026/Conference — Submitted to ICLR 2026_

### Official Review · Reviewer_DPfP · 2025-10-15

**Soundness:** 1
**Presentation:** 2
**Contribution:** 1
**Rating:** 2
**Confidence:** 3

**Summary:**

The paper studies aligned multi-objective optimization under convex assumptions.
The authors assume that all objectives share a common global minimizer and propose simple gradient-descent–type algorithms (PAMOO and MG-AMOO) that adapt the update direction based on the relative “gap” between each objective’s current value and its optimum.
They provide convergence guarantees in a newly defined Maximum-Gap metric, claiming an $\mathcal{O}(1/K)$ rate without requiring strong convexity. Section 6 extends the analysis to approximately aligned objectives, and a brief toy-function experiment is presented to illustrate convergence.
Overall, the paper is theoretically clean but focuses on a highly idealized problem setup with minimal empirical validation.

**Strengths:**

The theoretical analysis is mathematically sound, and the proofs appear correct.

**Weaknesses:**

The paper’s presentation is rather weak, and the fundamental assumption of the framework appears unrealistic.
In the introduction, the authors state that “the set of objectives is assumed to have a shared optimal solution” and then design gradient-descent algorithms under this assumption.
However, such a condition is almost never satisfied in practical multi-objective or multi-task learning settings—especially when objectives are derived from distinct data distributions, tasks, or losses.
Under this restrictive assumption, extending results from the non-convex to the convex case does not make the problem any closer to real-world relevance.

Although the authors attempt to generalize their theory in Section 6 by introducing an ε-aligned extension, this addition does not convincingly strengthen the practical relevance of the framework.\
The commonly adopted Pareto-front formulation in multi-objective optimization already naturally accommodates small deviations among optimal values; hence, the proposed “approximate alignment” assumption seems redundant.\
The authors should clearly justify why such a separate theoretical extension is necessary and provide a comparison with standard Pareto-based MOO algorithms under this relaxed condition.

Furthermore, the experimental section reinforces these concerns: the paper only provides synthetic toy-function demonstrations, without even a minimal empirical test on any machine-learning task.
This lack of connection between the theoretical framework and realistic ML optimization scenarios is concerning and casts serious doubt on the overall relevance of the work to the ML community.

**Questions:**

See weaknesses above

---

> ### Author Response · Authors · 2025-11-18
>
> Thank you for the feedback and recognizing the positive aspects of our work. We appreciate the detailed and useful feedback. We will address the weaknesses in their chronological order.
>
> On the realism of the AMOO and convex framework:
> Regarding the AMOO setting, we believe the framework is indeed realistic. Several practical scenarios naturally satisfy or closely approximate the shared optimum assumption.
>
> For instance, in using RL to enhance LLMs with reasoning abilities, multiple reward models are often considered. [1,2] examine both outcome- and process-based rewards, while [3,4] explore additional rewards such as accuracy, format adherence, response length, and math-specific rewards. Similarly, in RL-based training of text-to-image models, [5] employ four reward models—covering aesthetic quality, human preference, text-image alignment, and image sentiment—and report simultaneous improvements across all of them. Although [5] aim to find Pareto-optimal solutions, the empirical results indicate that the objectives are largely aligned.
>
> This aligns with a broader observation in RL noted by [6], where learning can be accelerated by leveraging multiple alternative reward formulations that yield the same optimal policy. This builds on prior findings that the choice of reward function (e.g., dense vs. sparse) can significantly influence training efficiency [7]. A parallel idea in statistics is that when labeled data is limited, closely related proxy tasks can be used to enhance predictive accuracy [8].
>
> More conceptual motivation stems from observations in multi-task learning, where multiple tasks are trained jointly with both shared and task-specific parameters. In several studies, improved performance across all tasks has been observed, suggesting that significant trade-offs between objectives may not always exist. Similar patterns have been reported in meta-learning, where models can rapidly adapt to new tasks with minimal additional training, and in reinforcement learning, where employing multiple reward functions helps better specify the policy or its representation.
>
> Following your comment, we will add an additional paragraph providing further motivation for our paper.
>
> Regarding convexity: while neural networks are indeed non-convex, most theoretical analyses of optimization algorithms used in practice, such as GD, ADAGRAD, and ADAM, provide convergence guarantees only under convexity. Therefore, analyzing the convex case remains a standard and informative starting point, as also stated in the introduction.
>
> On Section 6 (ε-aligned extension):
> The purpose of introducing the ε-aligned extension is to demonstrate the robustness of the algorithms when objectives are only approximately aligned. In our view, this generalization is conceptually important because it shows that convergence guarantees degrade smoothly as misalignment increases, which is a desirable stability property.
>
> Regarding the comparison to the Pareto-front formulation: our analysis focuses on convergence under the max-gap metric, whose solutions do not necessarily coincide with the Pareto front. For example, consider the simple convex functions
> f_i(x)=e^{(-1)^{i} x} , i=1,2, and  x∈R.
>
> These functions are 1-aligned with ε = 1. Both are convex, and their Pareto front spans the entire domain, whereas the max-gap solution is uniquely x=0. Hence, evaluating our results solely against the Pareto framework would not be an appropriate or fair comparison, as the objectives and optimality notions differ fundamentally.
>
> On the experimental evaluation:
> We agree that extended empirical validation on real ML tasks is a very important future direction (as it is written in section 8). However, the scope of the current work is primarily theoretical. Specifically, we:
> Prove a surprising lower bound for the naïve EW objective.
> Introduce a novel meta-algorithmic framework.
> Provide analysis of convergence guarantees for two algorithms (ours and PAMOO) under various settings.
> Show robustness of our method under approximate alignment.
> Include synthetic experiments to validate and illustrate these theoretical findings.
> We view these experiments as a conceptual proof-of-feasibility, while large-scale ML testing is left for future work building on this theoretical foundation.
>
> [1] Lightman et al. Let’s Verify Step by Step.
>
> [2] Uesato et al. Solving Math Word Problems with Process- and Outcome-Based Feedback.
>
> [3] Guo et al. DeepSeek-R1: Incentivizing Reasoning Capability in LLMs via Reinforcement Learning.
>
> [4] Kimi Team et al. Kimi K1.5: Scaling Reinforcement Learning with LLMs.
>
> [5] Lee et al. Parrot: Pareto-Optimal Multi-Reward Reinforcement Learning Framework for Text-to-Image Generation.
>
> [6] Dann et al. Reinforcement Learning Can Be More Efficient with Multiple Rewards.
>
> [7] Ng et al. Policy Invariance under Reward Transformations: Theory and Application to Reward Shaping.
>
> [8] Bastani. Predicting with Proxies: Transfer Learning in High Dimension.

---

> > ### Comment · Reviewer_DPfP · 2025-11-20
> >
> > Thank you for the timely and detailed response, and for the additional clarification regarding the theoretical aspects of the paper. The theoretical analysis itself is solid. My more conservative score stems from the difficulty of assessing the paper’s contribution to the broader machine learning community. Unfortunately, the current version of the paper—and the author response—does not fully resolve these concerns.
> >
> > First, I strongly encourage the authors to provide a clearer motivation and application background for the AMOO framework in the main paper. Even for theory-focused submissions, readers in a machine learning venue generally require a more explicit rationale for why a particular optimization setting is important or practically relevant.
> >
> > Second, my primary concern remains the strength of the core assumption. The response highlights several reinforcement learning scenarios where multiple rewards may appear aligned in practice. However, AMOO assumes that all objectives share the same global minimizer, which is a significantly stronger requirement and one that is rarely satisfied in realistic settings. For example, in using RL to enhance LLMs with reasoning abilities, response length is highly unlikely to share an optimal solution with other rewards; in text-to-image RL training, human preference models trained via DPO/PPO involve substantial biases that diverge sharply from the AMOO assumption. While theory does not need to mirror practice perfectly, the conceptual gap between the AMOO setting and these real-world optimization problems seems too large to ignore.
> >
> > From this perspective, the $\epsilon$-aligned extension introduced in the paper may actually have the potential to bridge this gap, because it appears much closer to the kinds of approximate alignment encountered in practice. At the moment, however, this extension is not discussed in depth. I believe emphasizing and significantly developing this part of the framework would strengthen the paper. Similarly, analyzing weaker mixed settings, where only some objectives share a minimizer, may also be a natural and more realistic direction.
> >
> > Finally, regarding experiments, I fully understand that a theoretical paper does not require extensive empirical evaluation. However, since the paper repeatedly emphasizes potential applicability to RL, even a very small-scale RL experiment would greatly help clarify the practical significance of the framework and make the contribution more tangible to the ML audience.
> >
> > For these reasons, I will need to maintain my original score. That said, I sincerely welcome continued discussion during the rebuttal period, and I am open to clarifying any of my comments if it helps strengthen the final version of the paper.

---

> > > ### Author Response · Authors · 2025-11-26
> > >
> > > We sincerely thank the reviewer for the detailed feedback and the thoughtful clarifications. We also appreciate the recognition of the theoretical significance of our work. However, we respectfully disagree with the implication that purely theoretical papers do not contribute to the broader machine learning community. In our view, theory and practice inform one another deeply, and progress in one domain often enhances understanding in the other. Below, we address the remaining points in the order they were raised.
> > >
> > > Regarding the request for clearer motivation:
> > > We will incorporate the motivational components mentioned in our previous response, as we committed to in our initial reply. Thank you.
> > >
> > > Regarding the assumptions in the paper:
> > >  We fully agree that results based solely on the AMOO assumption are weaker than those that also include the ε-aligned setting. Indeed, this is precisely why we added a dedicated section analyzing the ε-aligned regime. The difference between the two settings is conceptually small, and we chose to begin with the simpler and more accessible AMOO case before presenting the more general ε-aligned framework. This decision was made in order to present the material in the clearest and most accessible way. We understand that this may be a matter of taste, and we do not claim that our choice is the only correct one. That said, we would greatly appreciate it if you could kindly reconsider how this aspect influences your score.
> > >
> > > Regarding mixed settings:
> > > We agree that analyzing situations where only some objectives share a minimizer is a very interesting direction for future work. Currently, however, such cases fall outside the scope of this paper. We thank the reviewer for raising this insightful idea.
> > >
> > > Regarding the experiments:
> > > We fully agree that large-scale experiments can be valuable in illustrating the empirical implications of a theoretically grounded framework. However, we do not share the reviewer’s concern regarding the level of experimentation in this paper. The scope of this work is fundamentally theoretical, and in addition to standard convergence proofs, we also present a nontrivial lower bound for a naïve approach. A complete summary of the paper’s contributions appears in the final paragraph of our initial response. Therefore, in our view, the current level of experimental evaluation is appropriate for the scope and content of the paper, and does not detract from its theoretical nature.
> > >
> > > We believe that we have addressed all of the reviewer’s comments clearly and substantively, and we would be grateful if you could kindly reconsider your score in light of our clarifications.

---

### Official Review · Reviewer_UXW4 · 2025-10-20

**Soundness:** 3
**Presentation:** 2
**Contribution:** 2
**Rating:** 4
**Confidence:** 4

**Summary:**

This paper studies the convex multi-objective optimization problem. Compared with the existing work, this paper removes the strongly convex assumption and provide two methods with convergence guarantees.

**Strengths:**

1. Compared with existing studies, this paper adopts weaker assumptions.

2. The paper introduces a new metric, Maximum Gap (MG), to evaluate convergence.

3. It proposes two methods to address the formulated problem, which, compared with the EW baseline, eliminate the constant term in the convergence rate.

**Weaknesses:**

1. The writing and presentation can be improved. For example, the symbol $\lambda$ in line 132 is used before being defined, and Lemma 3, although presented in the appendix, is referenced in the main text without any brief introduction or explanation.

2. The main concern of this paper lies in its problem formulation. The paper focuses on a non-conflicting setting in multi-objective optimization (MOO). However, gradient conflict is one of the most challenging aspects of MOO and typically requires non-trivial efforts to address, as seen in algorithms such as MGDA. Moreover, general MOO aims to approximate the entire Pareto set, whereas this paper’s results are limited to specific points. Although the non-conflicting setting significantly simplifies the general MOO problem, the motivation for studying this particular case is unclear. Are there theoretical or empirical studies on this setting besides Efroni et al. (2025)?

3.  While the non-conflicting setting simplifies the general MOO problem—and as shown in Theorem 1, the baseline EW algorithm can already solve it, and the proposed PAMOO and MG-AMOO methods improve the upper bound only by a constant factor. However, these methods require access to the optimal function value for each individual objective. Is it practical to assume such information is available, and is it necessary to obtain it merely to achieve an improvement by a constant factor, especially when the number of objectives
m is typically small?

**Questions:**

See weaknesses.

---

> ### Author Response · Authors · 2025-11-18
>
> Thank you for the feedback and recognizing the positive aspects of our work. We appreciate the detailed and useful feedback. We will address the weaknesses in their chronological order.
>
> Regarding Lemma 3: You are correct — we will add a brief explanation of Lemma 3 in the main text to provide proper context before referencing it.
>
> Regarding the λ symbol: In line 132, we refer to the “strong convexity parameter,” which is conventionally understood as the minimal eigenvalue of the Hessian. Nevertheless, we agree that the definition should be made explicit, and we will revise the text to clearly state that λ denotes this eigenvalue.
>
> Regarding the motivation of AMOO setting, we believe the framework is indeed realistic. Several practical scenarios naturally satisfy or closely approximate the shared optimum assumption.
>
> For instance, in using RL to enhance LLMs with reasoning abilities, multiple reward models are often considered. [1,2] examine both outcome- and process-based rewards, while [3,4] explore additional rewards such as accuracy, format adherence, response length, and math-specific rewards. Similarly, in RL-based training of text-to-image models, [5] employ four reward models—covering aesthetic quality, human preference, text-image alignment, and image sentiment—and report simultaneous improvements across all of them. Although [5] aim to find Pareto-optimal solutions, the empirical results indicate that the objectives are largely aligned.
>
> This aligns with a broader observation in RL noted by [6], where learning can be accelerated by leveraging multiple alternative reward formulations that yield the same optimal policy. This builds on prior findings that the choice of reward function (e.g., dense vs. sparse) can significantly influence training efficiency [7,8,9,10]. A parallel idea in statistics is that when labeled data is limited, closely related proxy tasks can be used to enhance predictive accuracy [11].
>
> More conceptual motivation stems from observations in multi-task learning, where multiple tasks are trained jointly with both shared and task-specific parameters. In several studies, improved performance across all tasks has been observed, suggesting that significant trade-offs between objectives may not always exist. Similar patterns have been reported in meta-learning, where models can rapidly adapt to new tasks with minimal additional training, and in reinforcement learning, where employing multiple reward functions helps better specify the policy or its representation.
>
> Following your comment, we will add an additional paragraph providing further motivation for our paper.
>
> Regarding the comparison with a conflict setting with the Pareto front metric: our analysis focuses on convergence under the max-gap metric, whose solutions do not necessarily coincide with the Pareto front. For example, consider the simple convex functions
> f_i(x)=e^{(-1)^{i} x} , i=1,2, and  x∈R.
>
> These functions are 1-aligned with ε = 1. Both are convex, and their Pareto front spans the entire domain, whereas the max-gap solution is uniquely x=0. Hence, evaluating our results solely against the Pareto framework would not be an appropriate or fair comparison, as the objectives and optimality notions differ fundamentally.
>
> In Theorem 1, we present a lower bound for the EW algorithm under the Polyak step size (which is known to be optimal in the single-objective case). The result shows that the MG metric for EW is lower-bounded by m, whereas our proposed algorithms do not depend on the number of objectives m.
> It is also worth noting that the Polyak method itself assumes access to the optimal function values, so this assumption is consistent with standard practice in the analysis of such algorithms.
>
> [1] Lightman et al. Let’s Verify Step by Step.
>
> [2] Uesato et al. Solving Math Word Problems with Process- and Outcome-Based Feedback.
>
> [3] Guo et al. DeepSeek-R1: Incentivizing Reasoning Capability in LLMs via Reinforcement Learning.
>
> [4] Kimi Team et al. Kimi K1.5: Scaling Reinforcement Learning with LLMs.
>
> [5] Lee et al. Parrot: Pareto-Optimal Multi-Reward Reinforcement Learning Framework for Text-to-Image Generation.
>
> [6] Dann et al. Reinforcement Learning Can Be More Efficient with Multiple Rewards.
>
> [7] Ng et al. Policy Invariance under Reward Transformations: Theory and Application to Reward Shaping.
>
> [8] Luo et al. Accelerating Reinforcement Learning for Reaching Using Continuous Curriculum Learning.
>
> [9] Wang et al. Dynamic Subgoal-Based Exploration via Bayesian Optimization.
>
> [10] Hu et al. Learning to Utilize Shaping Rewards: A New Approach of Reward Shaping.
>
> [11] Bastani. Predicting with Proxies: Transfer Learning in High Dimension.

---

### Official Review · Reviewer_LXsx · 2025-10-30

**Soundness:** 2
**Presentation:** 2
**Contribution:** 2
**Rating:** 2
**Confidence:** 3

**Summary:**

This paper introduces a method for multi-objective optimization that relaxes the strong convexity assumption, focusing on convex functions with conditions like Lipschitz continuity or smoothness. The authors propose the Maximum-Gap (MG) metric to evaluate convergence in cases where a common optimal solution does not exist. Two algorithms, PAMOO and MG-AMOO, are presented with new convergence guarantees in the convex setting. MG-AMOO reduces multi-objective optimization to single-objective optimization for efficiency. The paper includes experiments comparing these algorithms to existing methods, demonstrating their performance under various settings.

**Strengths:**

This paper provides a theoretical foundation to support its conclusions. The authors develop the mathematical framework for aligned multi-objective optimization.

**Weaknesses:**

**Question 1** The paper's central assumption that all objectives share a common minimizer seems very strong and rarely satisfied in realistic multi-objective learning . If there are domains where this assumption is reasonable, please document them with additional references and empirical evidence from realistic settings.

**Question 2** Much of the development, including the MG metric converging to zero and the need to know $\{f_i^{\star}\}$, critically depends on this alignment. While Section 6 discusses an $\varepsilon$-approximate variant, the empirical and theoretical treatment still assumes access to optimal values and does not fully characterize the non-aligned regime where $C^{\star}=\varnothing$. This limits applicability.

**Question 3** Multi-objective optimization becomes more challenging when objectives conflict, especially in non-convex settings. While some works [1-2] have addressed this in non-convex cases, this paper focuses solely on convex optimization under the assumption of non-conflicting objectives. This seems overly restrictive, as real-world problems often involve conflicting objectives and non-convex landscapes. Would it be possible to extend the analysis to non-convex settings, or justify why the convex case alone is sufficient ?
**Question 4** The experimental comparison is limited to EW and PAMOO. Please consider recent multi-task/MOO solvers: MGDA, GradNorm, PCGrad/Grad Surgery. Even if theoretical regimes differ, empirical comparisons matter for readers choosing optimizers.

[1]Three-way trade-off in multi-objective learning: Optimization, generalization and conflict-avoidance.

[2] Mitigating gradient bias in multi-objective learning: A provably convergent stochastic .

**Questions:**

And some minor issues:

1.Typos: alighned $\rightarrow$ aligned; suprising $\rightarrow$ surprising; analgous $\rightarrow$ analogous.

2. State explicitly how $\{f_i^{\star}\}$ are obtained in experiments (exactly known or estimated surrogates).

3. Briefly restate Eq. (3)'s inequality intuition in the main text (one line) to help readers not flipping to the appendix.

---

> ### Author Response · Authors · 2025-11-18
>
> Thank you for the feedback. We appreciate the detailed and useful feedback. We will address the weaknesses in their chronological order.
>
> ## Weaknesses:
> Q1. Regarding the motivation of AMOO setting, we believe the framework is indeed realistic. Several practical scenarios naturally satisfy or closely approximate the shared optimum assumption.
>
> For instance, in using RL to enhance LLMs with reasoning abilities, multiple reward models are often considered. [1,2] examine both outcome- and process-based rewards, while [3,4] explore additional rewards such as accuracy, format adherence, response length, and math-specific rewards. Similarly, in RL-based training of text-to-image models, [5] employ four reward models—covering aesthetic quality, human preference, text-image alignment, and image sentiment—and report simultaneous improvements across all of them. Although [5] aim to find Pareto-optimal solutions, the empirical results indicate that the objectives are largely aligned.
>
> This aligns with a broader observation in RL noted by [6], where learning can be accelerated by leveraging multiple alternative reward formulations that yield the same optimal policy. This builds on prior findings that the choice of reward function (e.g., dense vs. sparse) can significantly influence training efficiency [7,8,9]. A parallel idea in statistics is that when labeled data is limited, closely related proxy tasks can be used to enhance predictive accuracy [10].
>
> More conceptual motivation stems from observations in multi-task learning, where multiple tasks are trained jointly with both shared and task-specific parameters. In several studies, improved performance across all tasks has been observed, suggesting that significant trade-offs between objectives may not always exist. Similar patterns have been reported in meta-learning, where models can rapidly adapt to new tasks with minimal additional training, and in reinforcement learning, where employing multiple reward functions helps better specify the policy or its representation.
>
> Following your comment, we will add an additional paragraph providing further motivation for our paper.
>
> Q2. In Section 6, the set C*can indeed be empty — that is, the objectives are not necessarily aligned. While we assume that the values fi*are known in advance, this does not imply alignment. As shown in Theorem 4, under such non-aligned conditions, the MG metric does not converge to zero.
> Furthermore, in our experiments, Problem P1 corresponds to an aligned case, whereas P2 and P3 represent non-aligned scenarios.
>
> Q3. We agree that the conflicting-objective case is more challenging and typically leads to convergence toward the Pareto front and not the Max-Gap. However, this is not the setting we consider in our work — please refer to our response to Q1 for the motivation behind focusing on the non-conflicting case.
> Regarding convexity: while neural networks are indeed non-convex, most theoretical analyses of optimization algorithms used in practice, such as GD, ADAGRAD, and ADAM, provide convergence guarantees only under convexity. Therefore, analyzing the convex case remains a standard and informative starting point, as also stated in the introduction.
>
> Q4. Our setting fundamentally differs from the conflicting-objective case typically addressed by Pareto-based methods.
> Regarding the comparison to the Pareto-front formulation: our analysis focuses on convergence under the max-gap metric, whose solutions do not necessarily coincide with the Pareto front. For example, consider the simple convex functions
> f_i(x)=e^{(-1)^{i} x} , i=1,2, and  x∈R.
> These functions are 1-aligned with ε = 1. Both are convex, and their Pareto front spans the entire domain, whereas the max-gap solution is uniquely x=0. Hence, comparing our results directly to Pareto-based MOO solvers (such as MGDA, GradNorm, or PCGrad) would not be appropriate or fair, as the objectives, metrics, and notions of optimality differ fundamentally.
>
> ## Questions:
> Q1 & Q3: Thank you for the comments — we will fix these issues.
> Q2: As mentioned in the appendix, we set the value to 0.
>
> [1] Lightman et al. Let’s Verify Step by Step.
>
> [2] Uesato et al. Solving Math Word Problems with Process- and Outcome-Based Feedback.
>
> [3] Guo et al. DeepSeek-R1: Incentivizing Reasoning Capability in LLMs via Reinforcement Learning.
>
> [4] Kimi Team et al. Kimi K1.5: Scaling Reinforcement Learning with LLMs.
>
> [5] Lee et al. Parrot: Pareto-Optimal Multi-Reward Reinforcement Learning Framework for Text-to-Image Generation.
>
> [6] Dann et al. Reinforcement Learning Can Be More Efficient with Multiple Rewards.
>
> [7] Ng et al. Policy Invariance under Reward Transformations: Theory and Application to Reward Shaping.
>
> [8] Luo et al. Accelerating Reinforcement Learning for Reaching Using Continuous Curriculum Learning.
>
> [9] Wang et al. Dynamic Subgoal-Based Exploration via Bayesian Optimization.
>
> [10] Bastani. Predicting with Proxies: Transfer Learning in High Dimension.

---

### Official Review · Reviewer_K18H · 2025-11-01

**Soundness:** 3
**Presentation:** 3
**Contribution:** 3
**Rating:** 6
**Confidence:** 1

**Summary:**

While many machine learning tasks can be optimized together without conflict, the "Aligned Multi-Objective Optimization" (AMOO) framework used to analyze this previously relied on an unrealistic strong convexity assumption. This paper relaxes that requirement, studying AMOO under standard convexity and smoothness conditions that are far more common in modern deep learning. The authors develop new analytical tools and scalable gradient-descent algorithms for this setting, proving their convergence and demonstrating their superiority over naive equal-weighting methods.

**Strengths:**

1. Well-motivated problem and novel approach.
2. The improved theoretical results and detailed analysis demonstrate a significant improvement over previous methods.

**Weaknesses:**

1. Compared to the detailed theoretical results presented in this paper, the experimental verification is very weak, as experiments are only conducted on 2-layer neural networks. While this is helpful for verifying the algorithm's theoretical results, it is insufficient for validating the algorithm's performance in real-world training scenarios. I think it would be better to conduct experiments on multi-objective training tasks, such as multi-task learning or multi-objective recommendation system training.

2. Why did the author only compare EW instead of some gradient-based multi-objective optimization algorithms, such as MGDA and its variant, which also claim a certain convergence speed?

**Questions:**

See the weaknesses part.

---

> ### Author Response · Authors · 2025-11-18
>
> Thank you for your positive and useful feedback. We will address the weaknesses in their chronological order.
>
> 1. We agree that evaluating the proposed methods on large-scale, real-world multi-objective tasks is important. As noted in Section 8, such empirical validation is beyond the current scope and is left for future work. The present paper focuses on the theoretical analysis and foundational aspects of the AMOO framework.
> 2. Our setting fundamentally differs from the conflicting-objective case typically addressed by Pareto-based methods.
> Regarding the comparison to the Pareto-front formulation: our analysis focuses on convergence under the max-gap metric, whose solutions do not necessarily coincide with the Pareto front. For example, consider the simple convex functions
> f_i(x)=e^{(-1)^{i} x} , i=1,2, and  x∈R.
> These functions are 1-aligned with ε = 1. Both are convex, and their Pareto front spans the entire domain, whereas the max-gap solution is uniquely x=0. Hence, comparing our results directly to Pareto-based MOO solvers (such as MGDA, GradNorm, or PCGrad) would not be appropriate or fair, as the objectives, metrics, and notions of optimality differ fundamentally.

---

> > ### Comment · Reviewer_K18H · 2025-11-28
> >
> > Thanks for your reply. I maintain my score.

---

### Meta-Review · Area_Chair_AdK7 · 2026-01-05

**Summary:**

This paper develops a convex-theory framework for Aligned Multi-Objective Optimization (AMOO), introducing the Maximum-Gap (MG) metric and proposing two simple gradient-based methods (PAMOO and MG-AMOO) with convergence guarantees under weaker assumptions than prior AMOO work.

Overall sentiment is mixed-to-negative. Reviewer K18H is moderately positive (scores around “good” and slightly above threshold), Reviewer UXW4 is borderline, while other two reviewers recommend reject, citing limited practical relevance and weak empirical evidence.Reviewer LXsx and Reviewer DPfp agree the analysis is mathematically sound, yet emphasize that the core AMOO assumption—objectives sharing a common global minimizer—is strong and rarely satisfied in realistic multi-objective learning, where gradient conflict is central. Although the rebuttal provides motivation (e.g., multi-reward RL) and discusses ε-alignment, reviewers remain unconvinced that the paper sufficiently bridges to practical regimes, and the ε-aligned extension is not developed deeply enough to address these concerns.

Empirically, the evaluation is viewed as somewhat limited: experiments are largely synthetic/toy (e.g., shallow networks) and do not test the methods on representative multi-objective tasks (multi-task learning, recommendation, RL). Reviewer LXsx also note missing comparisons to standard Pareto-based gradient methods (e.g., MGDA/GradNorm/PCGrad), and the need to clarify how per-objective optima  $f_{i}^{*}$ are obtained. Despite the authors' reply emphasizing potential RL applications and promising extensions to mixed settings, I remain unconvinced, as the framework's core assumptions (e.g., perfect alignment) clash with RL's complex trade-offs, leaving a gap between theory and practice.

Given these fundamental issues in motivation, baselines, and experimental validation, the paper is recommended for rejection.

**Reviewer Concerns:**

**Addressed**：

The authors’ rebuttal clarifies several issues raised by reviewers, especially around motivation and positioning.

In response to concerns that AMOO is unrealistic, the authors provide additional examples where objectives may be largely aligned (e.g., multi-reward reinforcement learning and multi-task settings) and commit to strengthening the motivation section accordingly.

They also explain why their evaluation targets convergence under the max-gap metric rather than Pareto optimality, arguing that standard Pareto-based solvers (e.g., MGDA/GradNorm/PCGrad) optimize a fundamentally different notion of optimality and thus are not directly comparable. This conceptual distinction is clearly stated and helps contextualize the limited baseline set.

Finally, the rebuttal acknowledges the scope as primarily theoretical and reiterates that the toy experiments are intended as proof-of-feasibility rather than a full empirical study.

**Outstanding**：

Despite these clarifications, multiple core concerns remain unresolved and are likely decisive for the final recommendation.

First, the central assumption that objectives share a common global minimizer is still viewed as too restrictive for typical multi-objective ML problems; the provided practical examples are suggestive but not supported by concrete empirical evidence showing near-alignment in realistic benchmarks.

Second, the ε-aligned extension, while promising, is not developed in sufficient depth (either theoretically or experimentally) to convincingly bridge the gap between the idealized AMOO setting and real-world misaligned objectives.

Third, the experimental validation remains minimal: evaluations are limited to small synthetic/toy networks, with no results on representative multi-objective tasks (multi-task learning, recommendation, RL), and key implementation details (e.g., how per-objective optima ($f_{i}^{*}$) are obtained or approximated) remain insufficiently documented.

Overall, the rebuttal improves clarity but does not eliminate the primary concerns about practical relevance and empirical support.

**Reviewer Scores:**

**Reviewer K18H: (Score: 6  -> Est. unchanged)**:

Active in discussion. Maintained the original positive score.

**Reviewer LXsx: (Score:2  -> Est. 4)**:

The reviewer’s main objections centered on (i) the realism of the AMOO assumption, (ii) whether the MG metric meaningfully applies beyond strict alignment, and (iii) the lack of comparisons to Pareto-based MOO solvers.

In the rebuttal, the authors provide a substantially clearer motivation for studying AMOO, citing practical scenarios (e.g., multi-reward RL and multi-task settings) where objectives may be largely aligned, and they commit to adding this motivation explicitly in the paper. They also directly address the “non-aligned regime” concern by clarifying that the aligned set can be empty and that MG does not necessarily converge to zero when objectives are not aligned (as reflected in their theoretical discussion), which reduces the risk of overclaiming. Finally, they explain why Pareto-front methods (MGDA/GradNorm/PCGrad) are not a fair apples-to-apples comparison under the max-gap notion of optimality, since the objectives and convergence target differ fundamentally.While the empirical evaluation is still limited and the setting remains restrictive, these rebuttal clarifications resolve several key misunderstandings and meaningfully improve the perceived soundness and positioning of the contribution.

I therefore suggest raising the score to a borderline level (4).

**Reviewer UXW4: (Score:4  -> Est. unchanged)**:

Unchanged. Rebuttal did not address the fundamental flaw.

**Reviewer DPfP: (Score:2  -> Est. unchanged)**:

Active in discussion. Maintained the original positive score.

---

### Decision · Program_Chairs · 2026-01-26

Reject